# Impact of age, sex and surgery type on engagement with an online patient education and support platform developed for total hip and knee replacement patients

**Rebecca Martin** [1] *, **Natalie Clark** [1], **Paul Baker** [1,2,3]

**1** James Cook University Hospital, South Tees Hospitals NHS Foundation Trust, Middlesbrough, England,
**2** Teesside University, Middlesbrough, England, **3** York University, Heslington, York, England

* Rmartin37@qub.ac.uk

## Abstract

### Background

Patients should be active participants in the management of their condition and provided with appropriate information throughout their care pathway. We piloted an online digital platform (ODP) to deliver patient education and support (PES) for patients undergoing total hip (THR) and knee replacements (TKR). The aim of this study is to analyse the demographics of patients using the ODP, determine how and when they accessed the ODP and how different sexes and age groups interacted with the ODP.

### Methods

Demographics and program library logs for patients registered to the ODP between 21st September 2017 and 28th May 2020 was obtained. Associations between age, sex, type of surgery and engagement were assessed using statistical comparisons.

### Results

1195 patients were registered on the ODP of which 832 (69.6%) accessed their carepacs. Patients accessed the content within their carepacs a mean of 29.1 times, spending a mean total time of 83 minutes. There was greater engagement for patients with a THR carepac (75.5%) compared to TKR (63.8%) (p<0.001). There were no differences in the proportion of patients that accessed the ODP or the total time spent on the platform dependent upon age (p = 0.34). Females accessed the platform more than males (p = 0.03). The use of a computer to access the ODP increased as age increased, whereas the use of a phone was favoured by the younger age groups (p<0.001).

### Conclusion

An ODP providing information to patients regarding their surgery is effective and demonstrates high levels of patient engagement. An online resource such as this does not discriminate against age or sex in terms of accessibility and can be useful for information provision.

**Data Availability Statement:** All relevant data are within the paper and its Supporting Information files.

**Funding:** The authors received no specific funding for this work.

**Competing interests:** The authors have declared that no competing interests exist.

## Introduction

Patients listed for primary joint replacements should be provided with appropriate information that is tailored to their specific condition. This supports shared decision making and allows them to make informed choices about their care [1]. NICE guidelines recommend that information should be specific to the procedure that are being offered, should be delivered in a format that can be easily understood and should start at the first appointment and continue whenever it is needed throughout the care pathway [2,3]. This encourages patients to actively participate in their own care, and promotes self-management of their condition. This is something that healthcare workers should be striving to provide [2,3].

Health literacy, defined as the "capacity to obtain, interpret, and understand basic health information and services and the competence to use such information and services to enhance health" remains a significant issue [4,5]. Failure to understand their condition and the expected outcomes of treatment can increase the risk of patients developing complications and lead to poorer outcomes[6]. Providing individualised education materials, designed for their specific condition improves the patient's health literacy and reduces anxiety surrounding their condition [6,7]. Information delivery direct to the patient also eliminates the need for patients to obtain information themselves and therefore reduces the risk of the patient relying on possible misinformation. In addition, patient education pre-operatively has been shown to reduce patient's length of stay in hospital and reduce overall costs of care. It results in increased empowerment, improved adherence to the care plan, improved independence and overall satisfaction with the procedure and outcome [8].

A systematic review including 14 studies assessing internet-based patient education programs compared to the traditional forms of patient education, such as printed materials highlighted improved patient satisfaction and an increased patient knowledge when using the internet-based programs [9]. However, a limitation identified within these studies was the exclusion of patients who were deemed to be inexperienced with internet use. In routine practice, this could exclude a large proportion of older patients, particularly in the population receiving THR and TKR. The average age across the studies included within the systematic review was 56.3 years which is much younger than the average age of patient receiving a THR (69.4 years) [10] or TKR (70.1 years) in the UK [10]. Therefore, from this systematic review, it was not possible to conclude that an internet-based program would be appropriate and beneficial for all patients undergoing a joint replacement given that certain groups were excluded.

Since 2017, South Tees NHS Trust has used an online digital platform (ODP) to deliver and provide patient education and support (PES) to patients undergoing THR and TKR. A concern when introducing an ODP, was that it might discriminate against specific demographics of patients who may be less technologically literate. These concerns were identified by a small number of patients already using the ODP [11], that suggested older patients might have some apprehensions navigating an online platform in comparison to the comfort of printed materials. The following study therefore aimed to determine: 1) How engagement and time spent on the ODP varied depending on whether the patient was undergoing a THR or TKR; 2) How engagement and time spent on the platform varied depending on age and sex; 3) How the method and time of access on the platform varied depending on age and sex.

## Materials and methods

### Study population

This study was a retrospective cohort of a consecutive series of patients registered on to the ODP between 21st September 2017 and 28th May 2020 that received either a THR or TKR

carepac. All patients registered on the ODP were included and there were no specific exclusions applied to the study population.

## Intervention: Description of the Online Digital Platform (ODP) intervention

Our ODP was developed to with the aim of providing patients and their family members with information related to their planned elective hip or knee replacement procedure, in a digital multimedia format, commencing at the point of listing for surgery and continuing through out their care pathway. In many institutions this information is provided in a single pre-operative face to face 'surgery school' alongside a written booklet of information. However, the 'surgery school' is resource intensive and information transfer is limited by the time available for these sessions. Therefore, when redesigning our own patient education and support offering we opted to digitize all of our resources and provide them using an online digital platform.

The ODP uses the GoWellHealth platform and allows the clinical team to prescribe bundles of content (termed a carepac) over a given period of time. The format, structure, timing and duration of the carepac is determined by the clinical team based on the content uploaded within the ODP content 'library'. A number of different carepacs can be created that are individualized to the procedure and patient group. These carepacs can then be delivered to the patient digitally to ensure information is provided across the entirety of the patient's care pathway. Consent is taken to register patients on the platform at the point of listing for surgery. Once the patient is registered they can access their account from which they can interact with the prescribed daily content or view information in their library of content. The information provided includes educational materials about their condition and planned surgical procedure; what to expect before, during and after surgery; exercises to do before and after surgery to aid recovery; lifestyle advice, including information about weight management, diet and smoking cessation; and advice about maximizing function independence and quality of life before and after surgery. Interactive patient questionnaires including Patient Reported Outcome Measures, Health questionnaires, Occupational therapy questionnaires and follow questionnaire post-surgery are also provided.

The ODP content is created and updated regularly by the patient's multidisciplinary care team (surgeon, nursing staff, Occupational therapists, Physiotherapists, anaesthetists). Access to the ODP is offered at the point of listing for surgery and continues until 12 months after surgery with content added to the individuals account depending where they are on the surgical pathway. The ODP supports a variety of media types including audio, video, e-mail, interactive forms and PDF documents. It can be accessed through any internet ready device including phone, tablet or computer and is supported by both Apple and Android devices.

The ODP was chosen as information can be provided to patients in a safe, understandable and time dependent manner that is driven by the clinical multidisciplinary teams. It also offers a secure and confidential communication module, allowing the patient to directly message their clinical team if they have any queries or concerns (including sending of any photos) which the team can respond through the platform. All patient interactions with the ODP are recorded within a digital program library log.

## Outcomes measures

For each participant demographic (age and sex), operative (THR or TKR) and program library log data was extracted from the ODP. Engagement with the ODP was assessed using 4 metrics derived from the program library log: 1) the date and time the ODP was accessed to view content; 2) the total number of times the ODP was accessed by each individual; 3) the duration of each

access period; 4) the device type that was used to access the content. When analysing the duration of each access per item of content, the time was capped at a maximum of 30 minutes. This was to correct for instances when the patient accessing the information may have left the content open inadvertently, leading to outlying results. Each piece of content was designed to be short and concise, therefore not expected to take the patient longer than 30 minutes per item of content.

### Ethics

Verbal or written consent to register patients on the platform is taken when the patient is listed for surgery to satisfy local information governance policies and GDPR. Within this consent process patients also agree that data generated from the ODP can be used for the purpose of research and quality improvement. This project is registered as a service evaluation with the local Trust's research and development department (REF 061219NC—Evaluation of the Go Well Health platform).

### Statistical analysis

For the purpose of comparative analysis, the ages of the patients were categorised into 5 groups: ≤50, 51-60, 61-70, 71-80 and >80 years old.

Outcome data analysed dependent upon the distribution and type of the data. Graphical plots were used to assess time of access. Parametric data (continuous normally distributed) was analysed using a Student's t-test mean or ANOVA and results are reported using mean and standard deviation (SD). Non-parametric data (continuous non-normally distributed) was analysed using Mann-Whitney U test or Kruskall-Wallis test and results are reported using median and interquartile range (IQR). Chi-squared or Fishers test were used for categorical data analysis.

A p value of <0.05 selected to indicate statistical significance. No formal samples size calculation was undertaken as all patients meeting the inclusion criteria over the study time frame were included. Analysis was conducted using SPSS version 26 (International Business Machines Corporation®).

## Results

During the period of analysis, a total of 1195 patients were registered on to the ODP. The patients had an average age of 67.4 years (range 24 to 98 years), 558 (46.7%) were male and 637 (53.3%) were female. Overall, 971 (81.2%) of the 1195 patients that were registered activated their account with 832 (69.6%) of these patients subsequently accessing their account to view the content provided. Characteristics of the proportion of patients activating their account and subsequently accessing the content provided are reported in Table 1.

Across the study period the ODP was accessed a total of 24,222 times by 832 patients with a median of 15 (IQR 5 to 56) accesses per patient and a median of 38 minutes (IQR 17-104) spent on the platform per patient (Table 2). The distribution of the data was non-normal due to it being skewed by a proportion of patients that engaged heavily with the ODP, spending considerable time accessing multiple pieces of content. The time spent on the platform per access was also skewed with a median time per access of 3.0 minutes (IQR 2.0 to 4.5) and model access time of 1-2 minutes (Table 2 and Fig 1).

### Impact of hip and knee replacement upon engagement and time spent on the ODP

Of the 1195 patients, 585 patients were registered for a THR carepac and 610 were registered for a TKR carepac (Table 1). Significantly more patients undergoing THR activated their ODP

**Table 1. Summary of number of patients registered, activated and accessed dependent on type of surgery, sex and age.**

| | Number of patients registered | | Number of patients activating their ODP account | | P value | Number of patients accessing their ODP account 1 or more times | | P value |
|---|---|---|---|---|---|---|---|---|
| Total | 1195 | | 971 (81.1%) | | | 832 (69.6%) | | |
| | *THR* | *TKR* | *THR* | *TKR* | | *THR* | *TKR* | |
| **Total** | 585 | 610 | 486 (83.1%) | 485 (79.5%) | p = 0.03 | 443 (75.7%) | 389 (63.8%) | p<0.001 |
| **Sex** | | | | | | | | |
| **Male** | 263 | 295 | 215 (81.7%) | 235 (79.6%) | p = 0.59 | 195 (74.1%) | 191 (64.7%) | p = 0.04 |
| **Female** | 322 | 315 | 271 (84.1%) | 250 (79.4%) | p = 0.12 | 248 (77.0%) | 198 (62.9%) | p<0.001 |
| **Age group** | | | | | | | | |
| **≤50** | 66 | 46 | 53 (80.3%) | 34 (76.1%) | p = 0.49 | 48 (72.7%) | 27 (58.7%) | p = 0.15 |
| **51–60** | 88 | 82 | 68 (77.3%) | 68 (82.9%) | p = 0.44 | 60 (68.2%) | 53 (64.6%) | p = 0.63 |
| **61–70** | 194 | 214 | 171 (88.1%) | 173 (80.8%) | p = 0.06 | 158 (81.4%) | 147 (68.7%) | p = 0.004 |
| **71–80** | 178 | 211 | 149 (83.7%) | 165 (78.2%) | p = 0.04 | 136 (76.4%) | 126 (59.7%) | p<0.001 |
| **>80** | 59 | 57 | 101 (87.0%) | 45 (78.9%) | p = 0.19 | 41 (35.3%) | 36 (63.2%) | p<0.001 |

account when compared to patients undergoing a TKR (83.1% versus 79.5%, p = 0.03). The activation rate was not influenced by sex, however there was a relationship with age with higher rates of activation in patients aged 61-70 years (p = 0.06) and 71-80 years (p = 0.04) in the THR group being observed (Table 1). Similarly more THR patients accessed the program to view at least 1 piece of content compared to TKR patients (75.5% versus 63.8%, p<0.001) (Table 1). This relationship was observed for both male (p = 0.04) and female patients (p<0.001) and was again driven by higher rates in patients aged 61-70 years (p = 0.004) and 71-80 years (p<0.001) in the THR group (Table 1).

There was a difference in the number of times patients accessed the platform with patients with a THR carepac accessing the system a higher median number of times compared to TKR (18 versus 12, p<0.001) (Table 2). The median total time spent on the program was also higher for THR patients at 59 minutes (IQR 21 to 124) compared to 38 minutes (IQR 12 to 91) for TKR patients (p<0.001). The median time per access was similar for the THR and TKR groups (3.0 and 3.1 minutes respectively) (Table 2).

## Engagement and time spent on the platform depending on age and sex

There was no significant difference in the proportion of patients accessing the ODP dependent upon their age group (p = 0.10) or sex (p = 0.75). There were no observed differences in the

**Table 2. Summary of number of accesses, time spent on the platform and time per access dependent upon type of surgery, sex and age.**

| Category | Number of times accessed | Total time spent on platform (mins) | Time per access (mins) |
|---|---|---|---|
| **Total n = 832** | Mean 29.1 (SD 39.3)<br>Median 15 (IQR 5–56) | Mean 83 (SD 104)<br>Median 38 (IQR 17–104) | Mean 3.9 (SD 3.7)<br>Median 3.0 (IQR 2.0–4.5) |
| **Hip n = 443** | Median 18 (IQR 7–44) | Median 59 (IQR 21–124) | Median 3.0 (IQR 2.0–4.4) |
| **Knee n = 389** | Median 12 (IQR 3–31) | Median 38 (IQR 12–91) | Median 3.1 (IQR 1.0–4.7) |
| **Male n = 386** | Median 13 (IQR 4–33) | Median 42 (IQR 16–101) | Median 3.1 (IQR 1.9–4.7) |
| **Female n = 446** | Median 17 (IQR 5–38) | Median 54 (IQR 18–118) | Median 3.0 (IQR 2.0–4.2) |
| **≤50 n = 75** | Median 19 (IQR 5–48) | Median 41 (IQR 17–105) | Median 2.2 (IQR 1.3–3.3) |
| **51–60 n = 113** | Median 18 (IQR 4–37) | Median 55 (IQR 16–107) | Median 2.8 (IQR 2.0–4.1) |
| **61–70 n = 305** | Median 16 (IQR 5–39) | Median 44 (IQR 16–117) | Median 3.0 (IQR 2.0–4.2) |
| **71–80 n = 262** | Median 12 (IQR 4–32) | Median 52 (IQR 18–115) | Median 3.4 (IQR 2.2–5.1) |
| **>80 n = 77** | Median 13 (IQR 5–26) | Median 50 (IQR 14–80) | Median 3.3 (IQR 2.1–5.0) |

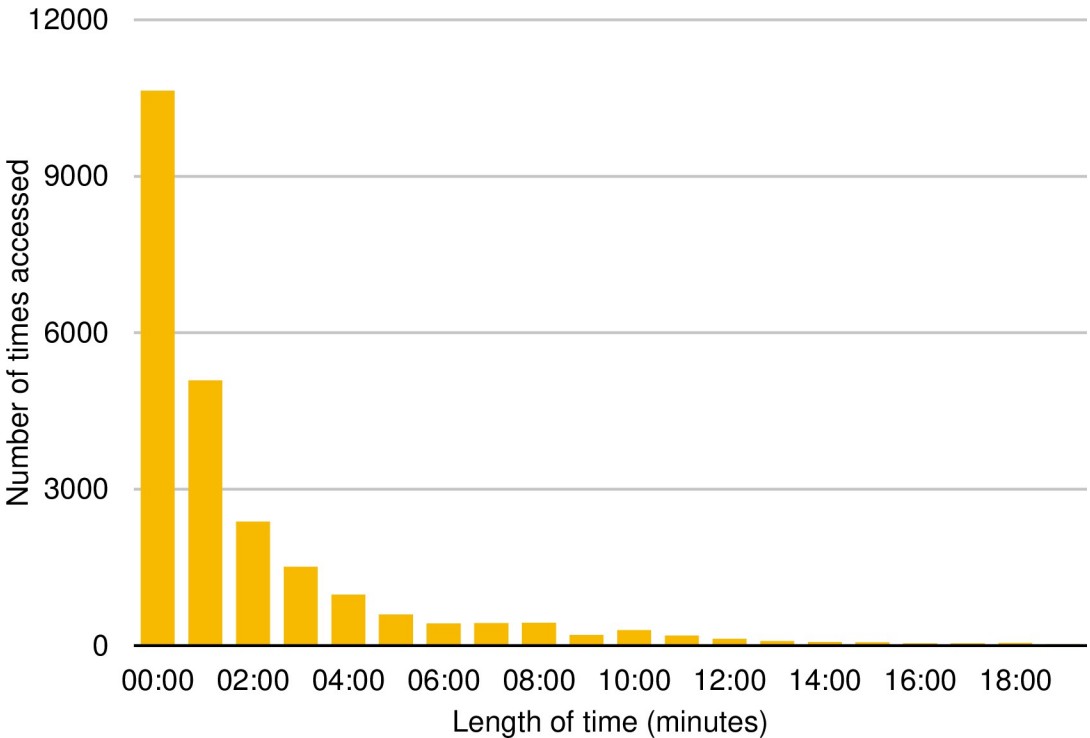

**Fig 1. Access times for each interaction with the ODP.**

number of times patients accessed the ODP (p = 0.19) or the total time spent on the platform dependent upon their age group (p = 0.34) (Table 2). However, the two oldest age groups (71-80 years p<0.001; >80 years p = 0.008) spent significantly longer on the program per access when compared with patients aged ≤50 years (Table 2). Patients aged ≤50 years spent a median time of 2.2 minutes on the program per access whereas patients 71-80 years and >80 years spent a median of 3.4 and 3.3 minutes on the program per access respectively.

When analysing the impact of sex we found that females accessed the program significantly more than males (p = 0.03). Female patients accessed a median of 17 times (IQR 5 to 38) whereas males accessed a median of 13 times (IQR 4 to 33) (Table 2). However, there was no difference in the median total time spent on the program between males and females (p = 0.08).

## Method and time of access on the platform depending on age and sex

All three devices that could be used to access the carepacs (computer, tablet, phone) were used by individuals in each of the age groups to varying degrees. The use of a computer increased in prevalence as age increased (36.8% in ≤50 years compared to 63.7% in >80 years) (Fig 2). In contrast the use of a phone to access the program was favoured by the younger age groups (39.8% in ≤50 years compared to 19.9% in >80 years). This observed difference in device use was statistically significant (p<0.001). The use of a tablet remained consistent throughout age groups ≤50 years to 71-80 with a decrease of use within the >80 age group.

The times of day that both programs were used were similar in distribution. There was variation in use across the 24-hour period (Fig 3) with a peak usage between 10 am—11 am. The was little activity between 1 am—6 am. All age groups accessed the program more frequently during the day typically from mid-morning until early evening although each individual age group had patients which accessed it outside of these peak times (Fig 4).

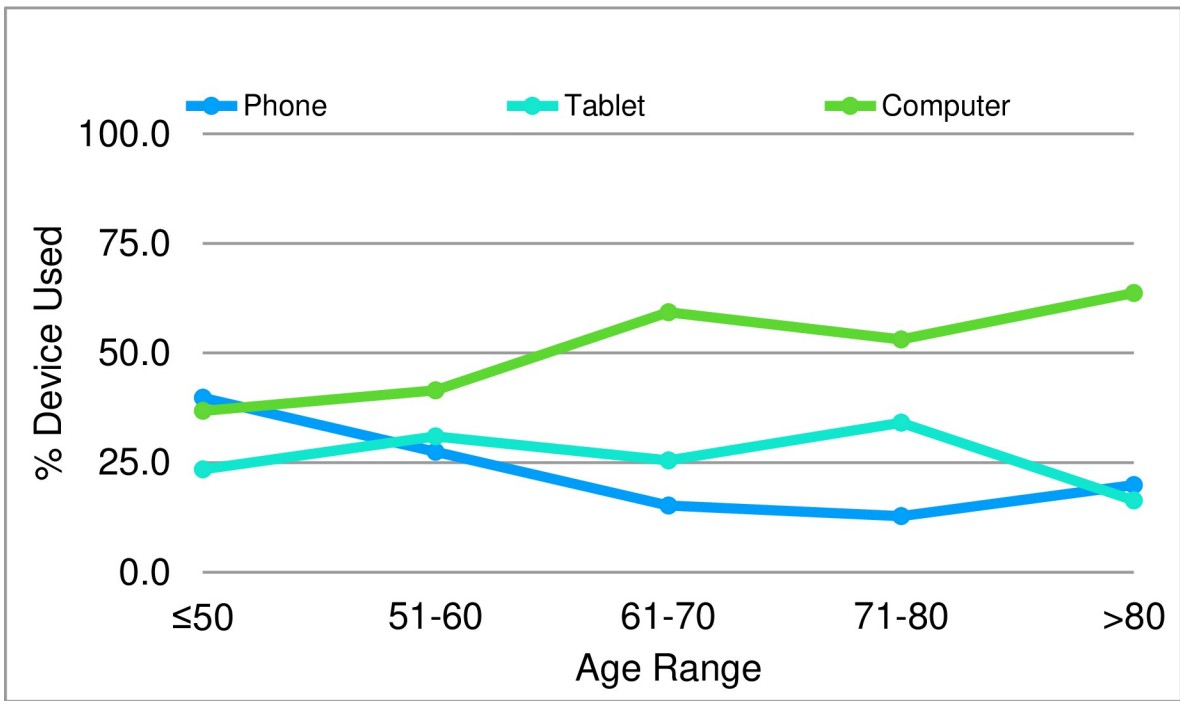

**Fig 2. Type of device used to access the ODP dependent upon age group.**

## Discussion

This study demonstrates that an ODP for the delivery of PES can achieve high levels of patient engagement and activity. Furthermore, this method of information delivery does not seem to discriminate against patients based on their age and sex as we observed excellent engagement and interaction irrespective of these factors.

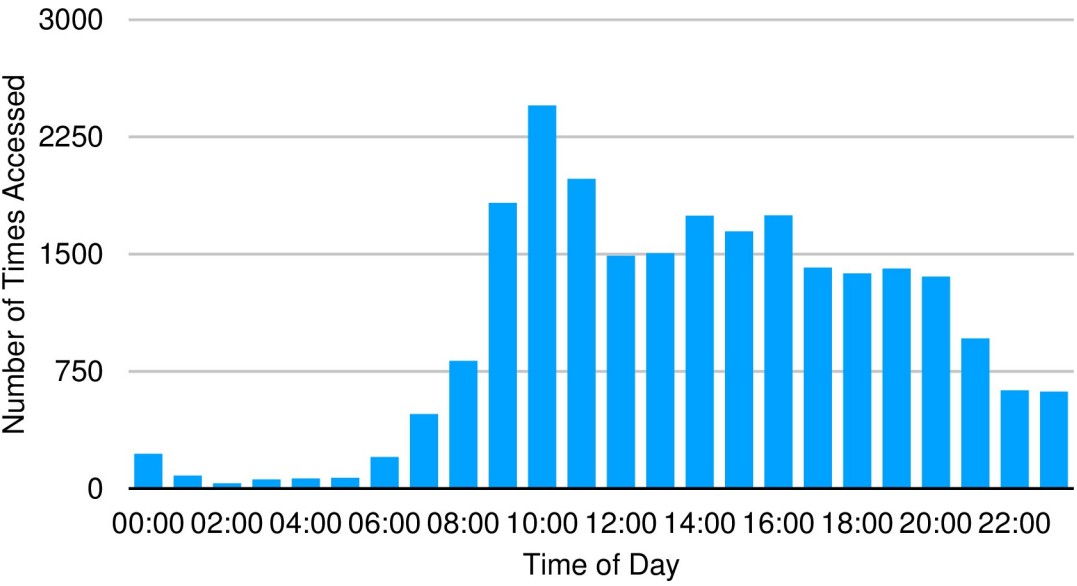

**Fig 3. Number of accesses to the ODP across the day.**

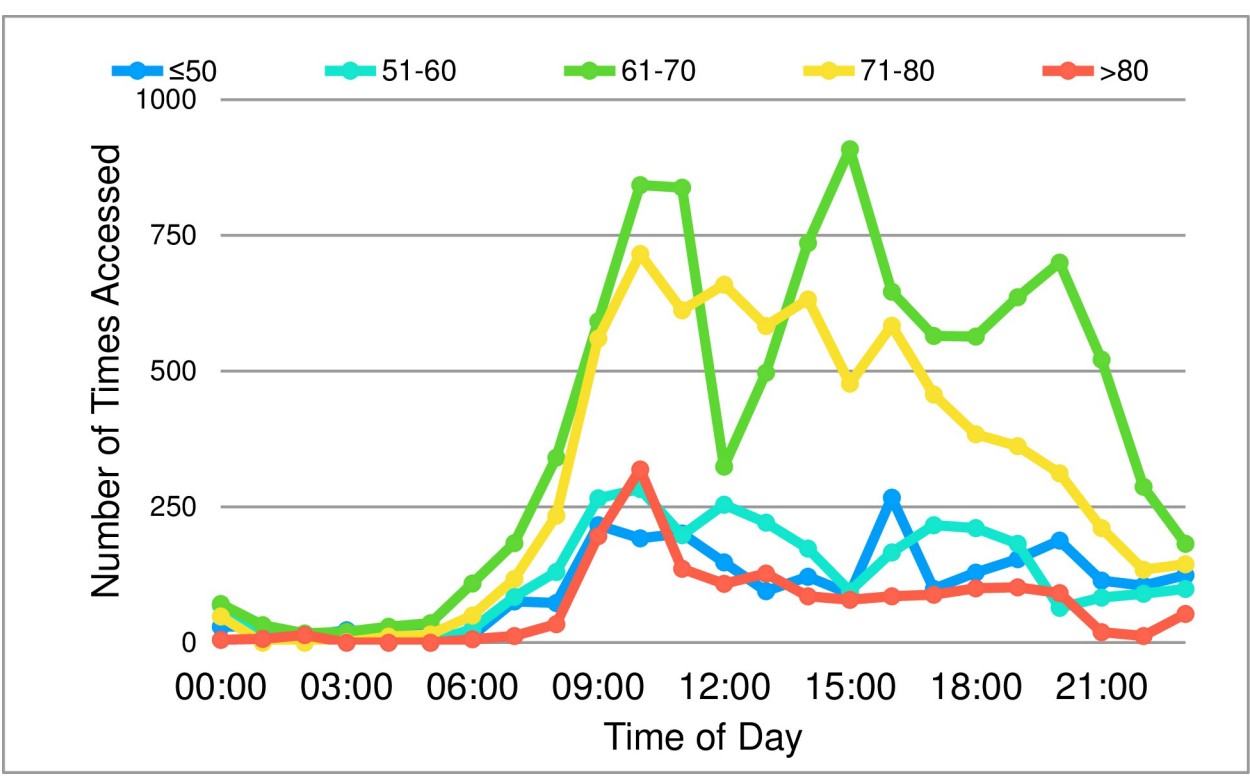

**Fig 4. Number of accesses to the ODP across the day dependent upon age group.**

This study included the entire patient population who had accessed the ODP and, as demonstrated within our results, captured a wide age range and sex distribution that is representative of the wider population of patients undergoing these procedures [12]. We found that the different age groups engaged with the program to a similar extent and for similar total lengths of time. The finding that there was no significant difference in rate of engagement for the older population compared to the younger groups is a new and reassuring finding as it addresses previous patient concerns for the older generation and such technological advancements [11].

Health literacy has a big impact on a patient's understanding and management of their condition and is said to be a predictor of a patients reported state of health [13]. Written information on health conditions and procedures needs to be tailored to the patient's reading level and there has been criticism that the format and content of previous patient facing information is too complex and above their level of health literacy [5]. A patient's understanding can be influenced by their literacy levels, understanding of medical terminology and understanding of the healthcare system. The ODP we have developed has helped us to address this possible disadvantage by providing information in a variety of formats (written, video, audio, interactive forms). It is reassuring that levels of activation (>80%) and interaction (70%) are high, coupled with a high median number of accesses which when combined suggest that patients are finding content on the ODP that they can engage with and that keeps them coming back.

A study published in 2017 [14] looked into the different modes of education delivery that patients preferred and it found that patients liked a variety of different formats including both written information and videos such as is being delivered via our ODP. An area of concern highlighted in this study was patient education regarding pain management and dealing with this after discharge from hospital. Our internet-based program allows patients to access the

necessary information easily from home and provides support, especially during the immediate period after discharge from hospital when pain is a reported issue. This also fits with the Department of Health long term aims to deliver more care at home via digital solutions [15].

The material included within the ODP is created by the medical team and can be updated quickly and efficiently as required. It allows the language used to be tailored to an understandable level or altered if feedback suggests otherwise. Patients also have the option of video and audio if preferred over the written media. The high uptake of usage of the program and the consistent usage across all groups of patients has demonstrated that the information is accessible and useful. There is an abundance of information regarding THR and TKR available on the internet but by providing a specific platform containing this information, we can reduce the risk of patients accessing information which is inaccurate or misleading. The volume of information provided on the platform is vast and too much to be feasibly covered during a clinic appointment. An advantage of the ODP is that it allows the patient to browse the information at their own pace and re-visit any topics they wish, patients have reported that this reinforced the information that was discussed at clinic appointments [11].

During the coronavirus pandemic many face to face clinic appointments were cancelled emphasising that our reliance on clinic appointments to give information is not always feasible. Patients who had their appointments cancelled included those who were immediately post-surgery but the ODP provided them with information on exercises and what to expect post-operatively and also provided a way to contact their team with any issues or concerns. These remotely accessed information sources are becoming more relevant due to the difficulty maintaining face to face appointments as well as ever-increasing time constraints in clinic. Providing a resource such as this means that potentially vulnerable people can browse the information in their own time from the safety of their home.

The patients who used the platform used it on their phones, tablets and computers. The older age category of patients preferred to use a computer to access the platform. If a program was designed that relied on smartphone-based applications, rather than a server that can be accessed across any device, this could potentially impact on certain patients' engagement and usage of the program, especially more elderly patients. This is supported by a study carried out in Australia which looked at cardiac education provided via mobile technology and found that the youngest age group (<56 years) was 4 times more likely to engage with mobile based technology and 5 times more likely to engage with information via a mobile application when compared with the oldest age group (69 years) [16]. Information such as this supports that a smartphone only platform would not be beneficial across all patient groups.

## Limitations

A number of patients who were registered to the ODP did not activate their accounts and, of those who did activate their accounts, a proportion did not access the content provided. The present study did not aim to understand why these patients did not activate their accounts or access the content as this could be due to a number of reasons (e.g. incorrect email provided). Patients are offered and provided with access to the ODP, it is then their responsibility to engage with the content provided and actively participate in their own care. Future studies could aim to evaluate these passive participants in order to fully understand the value of implementing an ODP to deliver PES for patients undergoing THR and TKR.

## Conclusion

A web-based ODP to deliver patient education and support demonstrates high levels of patient engagement irrespective of the patient's age and sex. This method of information delivery is

adaptable to individual patient requirements and can be helpful for the remote management of potentially vulnerable patient groups. A web-based ODP may be preferable over an app-based PES given the reliance on a computer to access the program in the older patient groups.

## Supporting information

**S1 File.**
(XLSX)

**S2 File.**
(XLSX)

## Author Contributions

**Formal analysis:** Rebecca Martin, Natalie Clark.

**Investigation:** Rebecca Martin, Natalie Clark, Paul Baker.

**Methodology:** Natalie Clark, Paul Baker.

**Supervision:** Paul Baker.

**Validation:** Paul Baker.

**Writing – original draft:** Rebecca Martin.

**Writing – review & editing:** Natalie Clark, Paul Baker.

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
