## [Decision Letter · Decision Letter 0]

6 Dec 2021

PONE-D-21-03212Impact of age, gender and surgery type on engagement with an online patient education and support program developed for hip  and knee replacement patientsPLOS ONE

Dear Dr. Martin,

Thank you for submitting your manuscript to PLOS ONE. After careful consideration, we feel that it has merit but does not fully meet PLOS ONE’s publication criteria as it currently stands. Therefore, we invite you to submit a revised version of the manuscript that addresses the points raised during the review process.

We look forward to receiving your revised manuscript.

Kind regards,

Marie-Pascale Pomey

Academic Editor

PLOS ONE

Journal Requirements:

2. Please include your tables as part of your main manuscript and remove the individual files. Please note that supplementary tables (should remain/ be uploaded) as separate "supporting information" files"

3. For more information on PLOS ONE's expectations for statistical reporting, please see https://journals.plos.org/plosone/s/submission-guidelines.#loc-statistical-reporting. Please update your Methods and Results sections accordingly.

Reviewers' comments:

Reviewer's Responses to Questions

**Comments to the Author**

1. Is the manuscript technically sound, and do the data support the conclusions?

Reviewer #1: No

2. Has the statistical analysis been performed appropriately and rigorously? 

Reviewer #1: No

3. Have the authors made all data underlying the findings in their manuscript fully available?

Reviewer #1: Yes

4. Is the manuscript presented in an intelligible fashion and written in standard English?

Reviewer #1: Yes

5. Review Comments to the Author

Reviewer #1: Intervention: This is a study investigating the use of an Internet-based patient education and support program for patients having hip and knee joint replacement. The patient education and support program is briefly described in the introduction and in the methodology section, but no details are given to what specific educational information is given to patients. This information should be provided using something such as the TIDIER Framework, description of evidence informing the educational materials, and readability level. This is particularly important given the readers do not have access to the program. OR is it publicly available? Having more details on the program is important for readers to appraise the study results.

Background: The introduction does not provide the background and current state of the literature on patient education and support program. As well, it does not highlight the importance of conducting this study. The introduction is focused on the authors’ specific situation and does not acknowledge previous research done. The introduction should be revised to focus on previous research in this area to highlight knowledge gap. This would provide the justification needed to understand the importance of conducting this study.

Methods: The authors should revise the objectives to make them clearer and more specific (e.g., to specify population of interest). There are minimal methods included – what is the study design and research methods used? It appears to be more of a quality improvement type initiative rather than a study. Results in the abstract related to age appears to be contradictory in the 2 sentences. In the conclusion, the intervention is described as being effective. How as effectiveness measured? From my understanding of the results, it was only access that was measured.

Results: The result section is difficult to understand and follow. The authors should revised to make it clearer. Furthermore, authors should provide a table of the statistical test conducted with the results (e.g., T-Tests and results). For example, both mean and median is presented with a p value but it is unclear what statistical analysis was done to achieve the p value.

For access to the education program, results indicate that patients accessed it 29.1 times. Are you able to also add the time periods for when it was accessed? Was it over 1 month or over 6 months?

Of concern is that a significant number of patients never accessed the program. Was anything done to understand why or to help them gain the knowledge elsewhere?

The authors aim to 1) analyze demographic characteristics of patients using the PES, 2) determine how and when patients access the PES, and 3) investigate if there are differences in 1 and 2 based on age and gender. The discussion and conclusion states that the PES support high level of patient engagement and activity independent of age and gender. However, the reader does not see how the data collected is related to patient engagement and activity as the relationship is not clear. Patient engagement and activity were not specifically measured by the study. The abstract conclusion states it was effective but how was effectiveness measured?

The background indicates that patients should be actively involved in decisions about their care – how was this measured? Also it describes patients inclusion in decision making is important but this intervention was only provided after they were placed on the list for surgery. How did the intervention support decision making?

The narrative uses gender but the results only report on sex (male, female) – how was gender measured? If it was based on male/female, then please change to sex.

The discussion is hard to follow as it includes a broad discussion of different topics related to PES that do not all appear to be of importance for this study. The authors should revised to make clear discussion points related to their specific findings. For example, start with a statement that links findings from this study with the broader literature discussed. Currently it discusses broader literature and the final sentence makes a link. I suggest the broader literature should be moved to the background and focus the discussion on discussing the results of this study as it relates to the literature. The last paragraph discusses the data but does not link to broader literature – what do people like to use from the broader literature – phones, tablets, or computers.

Conclusions – please remove ‘effectively engages’ or add evidence in the results that indicates how they were engaged and how effectiveness was measured.

No study limitations are provided – this is important to add.

Specific comments:

Line 9: Gender should not be written with an “s”.

Line 13. The authors should spell out the dates.

Line 23. Please give the number found instead of writing “p=NS”.

Lines 42-43. The authors should review the sentence as it is difficult to understand.

Line 52. Please specify overall satisfaction. Satisfaction with what?

Line 78. Patients are made aware of the program by which individual and how?

Line 102. The authors should spell out the dates.

Line 108. Why did a 30-minute cap was used? How long is interacting with the PES supposed to be? The readers need more information on that decision.

Line 114-119. This section is unclear making it difficult to clearly understand how the data will be statistically analyzed. The authors should revise the section.

Line 121. Informed consent should be discussed at the beginning of the methodology section.

Lines 133-135. Were patients characteristics similar between hip and knee replacement sample?

Line 135. Please modify “Range” to “range”.

Line 136. The total time access should be discussed with the average of time the program was access (lines 140-145).

Line 140. The authors should not describe patients are “hip patients” or “knee patients”. The authors should refer to the participants as “patient who underwent hip replacement” or something similar to this.

Line 142. Please be consistent across the manuscript when referring to PES (is “platform” referring to PES?).

Line 144. What is the difference between number of time access and interaction with the PES?

Lines 142-145. The reader does not understand why access to the program is described with mean (SD) and also with median (IQR).

Lines 138-149. This section is confusing for the reader making it difficult to follow. The authors should revised to make it clearer.

Line 159. “…the program per access when compared with patients aged <50 years. Patients aged ≤50 years…” Please be consistent with aged. Is it <50 years or ≤50 years?

Line 173. Please insert “statistically” before “significant”.

Lines 174-175. The authors should review the sentence as it is difficult to understand.

Lines 190-203. This should be in the introduction.

Lines 210-212. “The finding that there was no difference in the rate of engagement for the older population compared to the younger groups is a new and reassuring finding, particularly given the growth in these types of information platforms within orthopaedics.” Can the authors provide more information of why it is a “reassuring finding”.

Lines 225-231. This study did not measure health literacy of participants. As a result, “Having an online PES has helped us to combat this possible disadvantage.” should be revised as this is a strong conclusion not supported by study aims and results.

Figure 4 – YAY is in the title and it should be BY. Also the headings do not line up correctly in the PDF.

6. PLOS authors have the option to publish the peer review history of their article (what does this mean?). If published, this will include your full peer review and any attached files.

Reviewer #1: No

---

## [Author Response · Author response to Decision Letter 0]

20 Feb 2022

Marie-Pascale Pomey

Academic Editor

PLOS ONE

Date: 31st January 2022

Manuscript number: PONE-D-21-03212

Manuscript Title: Impact of age, gender and surgery type on engagement with an online patient education and support program developed for hip and knee replacement patients

Thank you for your time in reviewing our manuscript and providing us the opportunity to revise our work for reconsideration for your journal. The authors have considered each of the reviewers comments, please see the below responses in italics to each of the comments.

Journal Requirements:

Authors response: Manuscript formatted as per PLOS one guidelines 

2. Please include your tables as part of your main manuscript and remove the individual files. Please note that supplementary tables (should remain/be uploaded) as separate "supporting information" files"

Authors response: Tables have been included following the paragraph they are mentioned in. 

3. For more information on PLOS ONE's expectations for statistical reporting, please see https://journals.plos.org/plosone/s/submission-guidelines.#loc-statistical-reporting. Please update your Methods and Results sections accordingly.

Authors response: Statistical reporting updated as per the guidelines.

Authors response: Paul -Where can the minimal data set underlying the results be found?

Review comments to the Author

Intervention: This is a study investigating the use of an Internet-based patient education and support program for patients having hip and knee joint replacement. The patient education and support program is briefly described in the introduction and in the methodology section, but no details are given to what specific educational information is given to patients. This information should be provided using something such as the TIDIER Framework, description of evidence informing the educational materials, and readability level. This is particularly important given the readers do not have access to the program. OR is it publicly available? Having more details on the program is important for readers to appraise the study results.

Authors response: 

We have revised the “Materials and Methods” section to include more detail. Please see “Intervention: Description of the Online Digital Platform (ODP) Intervention”.

Background: The introduction does not provide the background and current state of the literature on patient education and support program. As well, it does not highlight the importance of conducting this study. The introduction is focused on the authors’ specific situation and does not acknowledge previous research done. The introduction should be revised to focus on previous research in this area to highlight knowledge gap. This would provide the justification needed to understand the importance of conducting this study.

Authors response: We have revised the first paragraph and added an additional reference relating to the NICE clinical guidelines. We have adjusted the introductory paragraph regarding health literacy which highlights why patient education and support was provided and why it is useful. The systematic review in the discussion has been moved to the introduction section to provide more literature.

Methods: The authors should revise the objectives to make them clearer and more specific (e.g., to specify population of interest). There are minimal methods included – what is the study design and research methods used? It appears to be more of a quality improvement type initiative rather than a study. Results in the abstract related to age appears to be contradictory in the 2 sentences. In the conclusion, the intervention is described as being effective. How as effectiveness measured? From my understanding of the results, it was only access that was measured.

Authors response: We have revised the three objectives at the end of the introduction section so they are more clear and specific and are in line with what have analysed in the results section. We have revised the conclusion and removed reference to “effectiveness”.

Results: The result section is difficult to understand and follow. The authors should revised to make it clearer. Furthermore, authors should provide a table of the statistical test conducted with the results (e.g., T-Tests and results). For example, both mean and median is presented with a p value but it is unclear what statistical analysis was done to achieve the p value.

Authors response: Table 1 has been amended to include the p values. The Results section has also been revised and should be clearer to follow. Mean value has been removed and has instead been discussed in terms of median and IQR as the data is skewed. 

For access to the education program, results indicate that patients accessed it 29.1 times. Are you able to also add the time periods for when it was accessed? Was it over 1 month or over 6 months?

Authors response: The time period that was analysed is as discussed in the methods section (21st September 2017 to 28th May 2020). We have reiterated this.

Of concern is that a significant number of patients never accessed the program. Was anything done to understand why or to help them gain the knowledge elsewhere?

Authors response: This has been added in as a limitation within the discussion section. 

The authors aim to 1) analyze demographic characteristics of patients using the PES, 2) determine how and when patients access the PES, and 3) investigate if there are differences in 1 and 2 based on age and gender. The discussion and conclusion states that the PES support high level of patient engagement and activity independent of age and gender. However, the reader does not see how the data collected is related to patient engagement and activity as the relationship is not clear. Patient engagement and activity were not specifically measured by the study. The abstract conclusion states it was effective but how was effectiveness measured?

Authors response: Revised the aims/objectives of the manuscript.

The background indicates that patients should be actively involved in decisions about their care – how was this measured? Also it describes patients inclusion in decision making is important but this intervention was only provided after they were placed on the list for surgery. How did the intervention support decision making?

Authors response: The background has been reworded to describe how patients should be actively involved in their care, this includes decision-making, active participation and self-management. Within this manuscript, we look at patient engagement (number of times accessed, how long they spent on the platform) with the platform which is how we have measured how actively involved patients have been with their own care.

Please see “Intervention: Description of the Online Digital Platform (ODP) Intervention” within the Materials and Methods section for a more in depth description of the intervention and how this supports decision making and when they are provided the content.

The narrative uses gender but the results only report on sex (male, female) – how was gender measured? If it was based on male/female, then please change to sex.

Authors response: All references to “gender” have been changed to “sex”.

The discussion is hard to follow as it includes a broad discussion of different topics related to PES that do not all appear to be of importance for this study. The authors should revised to make clear discussion points related to their specific findings. For example, start with a statement that links findings from this study with the broader literature discussed. Currently it discusses broader literature and the final sentence makes a link. I suggest the broader literature should be moved to the background and focus the discussion on discussing the results of this study as it relates to the literature. The last paragraph discusses the data but does not link to broader literature – what do people like to use from the broader literature – phones, tablets, or computers.

Authors response: We have reworked the discussion and have also added in some additional references (e.g. supportive literature for the use of phones, tablets and computers). 

Conclusions – please remove ‘effectively engages’ or add evidence in the results that indicates how they were engaged and how effectiveness was measured.

Authors response: Have removed “effectively engages” and reworded this to “demonstrates high levels of patient engagement”. Engagement was measured via how patients used the platform (e.g. time spent, number of times accessed) all of which have been analysed and reported within the manuscript.

No study limitations are provided – this is important to add.

Authors response: We have added a section for limitations and discussed those that did not activate their accounts or access the content.

Specific comments: 

Line 9: Gender should not be written with an “s”. 

Have changed all references to ‘gender’ to ‘sex’, the ‘s’ has automatically been removed as result.

Line 13. The authors should spell out the dates. 

Have spelled out the dates.

Line 23. Please give the number found instead of writing “p=NS”.

Number found has been given.

Lines 42-43. The authors should review the sentence as it is difficult to understand.

Sentence has been reviewed.

Line 52. Please specify overall satisfaction. Satisfaction with what?

Sentence has been reviewed.

Line 78. Patients are made aware of the program by which individual and how?

Clarified this is done at the patient’s initial consultation and by their treating surgeon.

Line 102. The authors should spell out the dates. 

Have spelled out the dates.

Line 108. Why did a 30-minute cap was used? How long is interacting with the PES supposed to be? The readers need more information on that decision.

This sentence has been elaborated.

Line 114-119. This section is unclear making it difficult to clearly understand how the data will be statistically analyzed. The authors should revise the section.

Revised, this should read clearer now.

Line 121. Informed consent should be discussed at the beginning of the methodology section.

Updated within the study methods section.

Lines 133-135. Were patients characteristics similar between hip and knee replacement sample?

Table one has been updated and now has total, THR and TKR each categorised into sex and age categories and also includes registered, activated and accessed. 

Line 135. Please modify “Range” to “range”.

Corrected.

Line 136. The total time access should be discussed with the average of time the program was access (lines 140-145).

Moved this sentence.

Line 140. The authors should not describe patients are “hip patients” or “knee patients”. The authors should refer to the participants as “patient who underwent hip replacement” or something similar to this.

Have re-referred to these patients consistently as assigned to “total hip replacement (THR) carepacs” or “total knee replacement (TKR) carepacs”. It is not possible to say underwent THR/TKR as some of these patients would not have had their operation yet and are viewing pre-operative information.

Line 142. Please be consistent across the manuscript when referring to PES (is “platform” referring to PES?).

Have referred to the platform as ‘online digital platform (ODP)’.

Line 144. What is the difference between number of time access and interaction with the PES?

The wording has been changed as we note access and interaction has been used interchangeably to mean the same thing. Interaction has been changed to accessed.

Lines 142-145. The reader does not understand why access to the program is described with mean (SD) and also with median (IQR).

Revised – discussed using median and IQR, removed mean.

Lines 138-149. This section is confusing for the reader making it difficult to follow. The authors should revised to make it clearer.

Revised this section.

Line 159. “…the program per access when compared with patients aged <50 years. Patients aged ≤50 years…” Please be consistent with aged. Is it <50 years or ≤50 years?

Corrected.

Line 173. Please insert “statistically” before “significant”.

Inserted.

Lines 174-175. The authors should review the sentence as it is difficult to understand.

Reworded.

Lines 190-203. This should be in the introduction.

Moved to the second paragraph in the introduction.

Lines 210-212. “The finding that there was no difference in the rate of engagement for the older population compared to the younger groups is a new and reassuring finding, particularly given the growth in these types of information platforms within orthopaedics.” Can the authors provide more information of why it is a “reassuring finding”.

Reinforced with previous qualitative research on our online digital platform.

Lines 225-231. This study did not measure health literacy of participants. As a result, “Having an online PES has helped us to combat this possible disadvantage.” should be revised as this is a strong conclusion not supported by study aims and results.

Health literacy paragraph revised.

Figure 4 – YAY is in the title and it should be BY. Also the headings do not line up correctly in the PDF.

Corrected.

All of the above amendments have been reflected in the manuscript. We welcome additional comments if any. 

Thank you for your time in reconsidering our manuscript and we look forward to your response.

Sincerely, 

Rebecca Martin, Natalie Clark and Paul Baker

Secretary to Prof Baker Department of Trauma and Orthopaedics,

James Cook University Hospital,

Marton Road,

Middlesbrough,

TS4 3BW

Email: rmartin37@qub.ac.uk

Telephone: 07889005185

---

## [Editor Report · Decision Letter 1]

1 May 2022

PONE-D-21-03212R1Impact of age, sex and surgery type on engagement with an online patient education and support platform developed for total hip and knee replacement patientsPLOS ONE

Dear Dr. Martin,

Thank you for submitting your manuscript to PLOS ONE. After careful consideration, we feel that it has merit but does not fully meet PLOS ONE’s publication criteria as it currently stands. Therefore, we invite you to submit a revised version of the manuscript that addresses the points raised during the review process.

We look forward to receiving your revised manuscript.

Kind regards,

Marie-Pascale Pomey

Academic Editor

PLOS ONE

Journal Requirements:

Additional Editor Comments (if provided):

First reviewer: Point 4.

defines a study's minimal data set as the underlying data used to reach the conclusions drawn in the manuscript and any additional data required to replicate the reported study findings in their entirety. All PLOS journals require that the minimal data set be made fully available. For more information about our data policy, please see http://journals.plos.org/plosone/s/data-availability.

Authors response: Paul -Where can the minimal data set underlying the results be found?

Can you answer to the question ?

---

## [Author Response · Author response to Decision Letter 1]

10 May 2022

A minimal data set for hip related data and knee related data can be found under supporting information.

---

## [Editor Report · Decision Letter 2]

30 May 2022

Impact of age, sex and surgery type on engagement with an online patient education and support platform developed for total hip and knee replacement patients

PONE-D-21-03212R2

Dear Dr. Martin,

We’re pleased to inform you that your manuscript has been judged scientifically suitable for publication and will be formally accepted for publication once it meets all outstanding technical requirements.

Kind regards,

Marie-Pascale Pomey

Academic Editor

PLOS ONE

---

## [Editor Report · Acceptance letter]

13 Jul 2022

PONE-D-21-03212R2 

Impact of age, sex and surgery type on engagement with an online patient education and support platform developed for total hip and knee replacement patients 

Dear Dr. Martin:

I'm pleased to inform you that your manuscript has been deemed suitable for publication in PLOS ONE. Congratulations! Your manuscript is now with our production department. 

Kind regards, 

on behalf of

Dr. Marie-Pascale Pomey 

Academic Editor

PLOS ONE